# Potential Effects of Leukotriene Receptor Antagonist Montelukast in Treatment of Neuroinflammation in Parkinson’s Disease

**DOI:** 10.3390/ijms22115606

**Published:** 2021-05-25

**Authors:** Johan Wallin, Per Svenningsson

**Affiliations:** Laboratory of Translational Neuropharmacology, Department of Clinical Neuroscience, Karolinska Institutet, 17177 Stockholm, Sweden; johan.wallin@ki.se

**Keywords:** montelukast, alpha-synuclein, leukotriene, microglia, neuroinflammation, Parkinson’s disease

## Abstract

Parkinson’s disease (PD) is a neurodegenerative disorder where misfolded alpha-synuclein-enriched aggregates called Lewy bodies are central in pathogenesis. No neuroprotective or disease-modifying treatments are currently available. Parkinson’s disease is considered a multifactorial disease and evidence from multiple patient studies and animal models has shown a significant immune component during the course of the disease, highlighting immunomodulation as a potential treatment strategy. The immune changes occur centrally, involving microglia and astrocytes but also peripherally with changes to the innate and adaptive immune system. Here, we review current understanding of different components of the PD immune response with a particular emphasis on the leukotriene pathway. We will also describe evidence of montelukast, a leukotriene receptor antagonist, as a possible anti-inflammatory treatment for PD.

## 1. Introduction

Parkinson’s disease (PD) is the second most common neurodegenerative disorder after Alzheimer’s disease (AD) [1]. Clinical manifestations of PD can vary, but a formal diagnosis relies on the presence of bradykinesia with rigidity and/or rest tremor according to Movement Disorder Society (MDS) criteria for PD [2]. Non-motor symptoms, such as hyposmia, constipation, depression, and rapid eye movement (REM) sleep behavior disorder, are common and can in many cases manifest before classical motor symptoms. In later years, more emphasis has been put on non-motor symptoms, especially in the early stages of PD and which is evident in the proposed prodromal PD criterion by MDS [3]. The cause of disease phenotype is principally a degeneration of dopaminergic neurons in substantia nigra pars compacta, but other areas of the central and peripheral nervous system are also affected. The pathogenesis is multifactorial but protein aggregates called Lewy bodies, mainly composed of misfolded α-synuclein, are believed to be the main cause of disease progression [4]. There is however growing evidence that immune responses and in particular increased microglial activity is a significant contributor to neurodegeneration in PD [5]. Current treatment strategies are focused on symptom relief. Drugs that enhance intracerebral dopamine concentrations or stimulate dopamine receptors are efficient, at least in the early stages, against motor symptoms. However, no neuroprotective or disease-modifying treatments are available [6].

This review will summarize aspects of the role of neuroinflammation in PD with a particular emphasis on microglial activation through the leukotriene signaling pathway. This is followed by a discussion on the possible use of the leukotriene receptor antagonist montelukast in the treatment for PD.

## 2. Neuroinflammation in PD

As in other parts of the body, inflammation plays a fundamental role in the central nervous system (CNS) to protect from damage and pathogens. During recent years, more evidence has been found to suggest that hyperactivation of innate and adaptive immune responses plays an important role in the progression of neurodegeneration [7]. There are still many areas of uncertainty such as which parts of the innate and immune system play prominent roles in pathology, what triggers neuroinflammation, and if it is possible to analyze the evolution of various forms of inflammation during the course of the disease [8].

### 2.1. Astrocytes

The main physiological features of astrocytes are centered around homeostatic function. They regulate pH, glutamate levels, provide metabolites, play a major role in neurogenesis, and regulate the blood-brain barrier [9].

In response to injury, astrocytes proliferate to the site and aid by isolating the lesioned area and facilitate regeneration of nervous tissue. Regulation is done mainly by danger-associated molecular patterns (DAMP) or pathogen-associated molecular patterns (PAMP) depending on whether the lesion is caused by a pathogen or other neuronal damage [10]. Astrocytes can be further divided into A1 and A2 astrocytes where A1 is believed to possess direct lethal effects on neurons and oligodendrocytes and A2 appears to possess neuroprotective abilities [11].

Astrocytes of the neurotoxic A1 phenotype secrete pro-inflammatory mediators, such as interleukin(IL)-1β and tumor necrosis factor (TNF)-α, thus further activating the inflammatory response in the CNS [12].

Evidence from genetic studies on PD-associated genes, such as Parkinsonism associated deglycase (*PARK7*), suggest these genes are implicated in astrocyte function and therefore points to possible connections between pathological astrocytes and PD [13]. In general, malfunctioning astrocytes are known to cause neuronal damage in a variety of CNS diseases but are believed to play a more significant role in AD, progressive supranuclear palsy, corticobasal syndrome, and amyotrophic lateral sclerosis than in PD [10,14].

### 2.2. Microglia

Microglia is the most abundant immune cell in the CNS. During physiological conditions, microglia supports a variety of functions such as synaptic pruning and remodeling [15]. Microglia has phagocytic properties and continuously monitors surrounding areas for brain injury, debris, and other damaged tissue. The monitoring happens in a resting state and upon a pathological trigger, microglia changes morphologically into the activated state [15]. The activated microglia accumulate around the trigger and continues repairing functions by engulfing debris and secreting pro-inflammatory factors [5]. These factors include cytokines and chemokines, such as IL-1β, IL-6, IL-12 TNF-α, and interferon (IFN)-γ, which further induce immune activity and can regulate neuronal survival [5]. Activation of microglia leads to upregulation of nitrous oxide synthase and consequently the production of free radicals which along with reactive oxygen species (ROS) can lead to oxidative damage to surrounding neurons [15]. Microglia has been studied extensively as an important factor in PD disease progression.

Evidence for elevated activation of microglia in vivo has been found through positron emission tomography (PET) where different tracers targeting translocator protein (TSPO), a mitochondrial membrane protein on activated microglia, shows increased binding in PD patients over healthy controls [16,17]. ^11^C-PK11195 is a first-generation TSPO tracer and has been used extensively to study microglial activation since it was synthesized in the 1980s [18]. ^11^C-PK11195 was used in a study on rapid eye movement (REM) sleep disorder patients and showed increased microglial activation in substantia nigra and a decrease of dopamine uptake compared to healthy controls [19]. First-generation TSPO tracers are limited by low specific binding affinity to different brain regions which led to the development of second-generation tracers such as ^11^C-PBR28 [20]. Binding specificity and signal-to-noise ratio were improved but clinical studies have shown mixed results. In a study with 16 PD patients and 16 healthy controls, ^11^C-PBR28 PET did not show a significant difference in microglial activation [21]. The less specific binding of first-generation tracers and subsequently increased uptake in non-relevant areas could in part explain why newer tracers do not show the same promising results.

In animal models of PD, induction of microglial activation through toxins has been shown to lead to loss of dopaminergic neurons in substantia nigra and striatum with worsening of both motor and cognitive symptoms [22,23].

### 2.3. Toll-Like Receptors

Toll-like receptors (TLR) are a family of transmembrane proteins key to the activation of the innate immune response. Toll-like receptors detect PAMPs or DAMPs. Recognition of either type of molecule activates a signaling cascade with upregulation of inflammatory mediators through nuclear factor (NF)-κB [24]. In particular, TLR2 and 4 have been shown to be upregulated in PD patients and animal models of PD [24]. α-synuclein has been shown to act as a DAMP and a knockout mouse model of microglia without TLR4 showed complete inhibition of α-synuclein phagocytosis [25,26]. Furthermore, complete knockout of TLR4 expression in a mouse model attenuated dopamine depletion, astrogliosis, and production of inflammatory mediators [27]. This highlights the varied effects of TLR4 activation throughout the CNS.

### 2.4. Autophagy

Autophagy and the autophagy-lysosomal pathway is a crucial intracellular process for the degradation of long-lived proteins and cellular organelles. Studies have shown inhibition of autophagy by abnormal α-synuclein with aggregation of these proteins as a result [7]. Aggregation further disrupts cell functions with activation of inflammatory pathways and cell death as a result [28]. Restoration of autophagy function in PD poses an attractive treatment target. Temsirolimus, an approved renal cell carcinoma drug, has been shown to elicit neuroprotective effects in an animal model of PD [29]. The study showed positive modulation of the autophagic process and reduction of astrocyte and microglia activation.

### 2.5. Monocytes

Monocytes are leukocytes derived from myelomonocytic stem cells in the bone marrow and are characterized by their ability to phagocytose, produce cytokines, and present antigen [30]. Monocytes share the same lineage as microglia and share many functional features such as phagocytosis. During inflammatory conditions, monocytes can enter the CNS and further activate inflammatory responses [31]. In a study on monocyte changes in plasma from PD patients, a higher concentration of classical monocytes and TLR4 positive monocytes were found compared to healthy controls. This shift in the monocyte population indicates a pro-inflammatory state [32]. Interestingly, increased expression of monocyte TLR4 was seen in a study on REM sleep behavior disorder patients and was correlated with an increase in striatal microglial activation and decrease in DOPA uptake in putamen as measured by PET [33]. Since isolated REM sleep behavior is regarded as a prodromal phenotype of synucleinopathies including PD and dementia with Lewy bodies, this suggests that peripheral immune changes are present in early disease stages.

### 2.6. Adaptive Immunity

The CNS has intrinsic properties that give it a general immune privilege. The blood-brain barrier restricts the transfer of peripheral immune components and there is no lymph system to accommodate an adaptive immune response within the CNS itself [12]. Peripheral immune cells in the form of T-cells and B-cells can be recruited although the exact mechanisms are not fully understood. It is however believed that microglia can act as an antigen-presenting cell and thus trigger a T-cell mediated response [34]. Antibodies against dopaminergic (DA) neurons have been identified in some PD patients, which gives evidence for the involvement of B-cells, but a PD-specific antigen has not been found and it is unlikely that PD pathology is driven by antibodies against a shared antigen [35].

Evidence for an elevated T-cell response in PD was first shown in post-mortem brain sections from PD patients [36]. T-cell infiltration in the substantia nigra and striatum has also been observed in 1-methyl-4-phenyl-1,2,3,6-tetrahydropyridine (MPTP) neurotoxin models of PD [34]. Further evidence of T-cell involvement was found in a study of human-induced pluripotent stem cells (hiPSCs) cultured into midbrain neurons from PD patients. The study found that T-cells, specifically Th17 derived CD4 T-cells, induced neuronal death through IL-17 signaling and activation of NF-κB. Increased frequency of IL-17 producing T-cells was also found in blood from PD patients [37]. A correlation was seen between IL-17 and non-motor scores where higher levels of IL-17A in plasma were correlated to more severe symptoms in a PD cohort [38].

If the adaptive immune response is an early cause of disease or a late consequence is debated but compelling evidence for an early T-cell response was found in a very interesting study by Lindestam and colleagues [39]. The study focused on α-synuclein-specific T-cell reactivity where reactivity was shown to be highest shortly after diagnosis in two independent cohorts of PD. In the same study, peripheral blood mononuclear cell (PBMC) samples spanning 20 years were available from one patient with PD. α-synuclein-specific T-cell reactivity was analyzed and reactivity was detectable 10 years before the onset of motor symptoms and PD diagnosis [39].

### 2.7. Cytokines

Cytokines are polypeptides secreted from immune and non-immune cells which play an important role in cell development and functions, such as regulation of apoptosis and inflammatory response. Cytokine is a broad term that includes chemokines, interferon, interleukins, and tumor necrosis factor. They act locally in a paracrine or autocrine fashion in most cases but some enter the bloodstream [40].

Cytokines are believed to play an important role in neurodegenerative and neuroinflammatory conditions. Evidence for elevated cytokine levels has been shown in two meta-analyses where one by Qin and colleagues found elevated peripheral plasma concentrations of IL-6, TNF-α, and IL-1β and the other by Chen and colleagues found elevated CSF concentrations of the same cytokines [41,42]. In a study by Green and colleagues, cytokines IL-6 and IL-17A were not shown to be elevated in PD compared to healthy controls on a group level but a correlation was seen between these cytokines and disease severity. IL-6 was correlated to the severity of motor symptoms and IL-17A was correlated to the severity of non-motor symptoms [38].

IL-1β is overly expressed in activated microglia but can also be secreted from other glial cells and neurons. It can exert toxic effects directly by binding to neurons or indirectly by further activating inflammatory pathways and induce reactive oxygen/nitrogen species in a positive feedback loop [43]. TNF-α has been shown to have similar effects as IL-1β but there is also some evidence that binding of TNF-α to DA neurons could trigger cell death directly through pro-apoptotic pathways [43,44]. Interestingly, in a retrospective study of inflammatory bowel disease (IBD), there was a substantially reduced PD incidence among IBD patients that were exposed early to anti-TNF-α therapy [45]. IL-6 can be produced by many brain resident cells, but the IL-6 receptor has only been found on microglia. The binding of IL-6 to microglia can trigger further activation of inflammatory pathways but its exact importance in PD is not yet known, although the correlation with motor symptoms warrants further research [38,46].

### 2.8. Genetic and Environmental Factors

Genetically, missense mutations in the leucine-rich repeat kinase 2 (*LRRK2*) gene are considered to be the greatest contributor to autosomal-dominant inherited PD [4]. It is a large protein involved in many cellular processes including autophagy, membrane transport, and synaptic development; *LRRK2* is abundant in monocytes and microglia. Interestingly, *LRRK2* mutation has been shown to affect many parts of the immune response such as inflammatory cytokine production and microglia activation [26,47]. Furthermore, susceptibility to PD has been shown to be associated with polymorphisms in the human leukocyte antigen (HLA) region which encodes vital antigen recognition and presentation proteins [48,49]. Evidence for a genetic association between adaptive immunity and PD has also been seen in whole genome sequencing where T-cell-specific immune annotations were enriched in PD patients as compared with healthy controls [50]. Genetic association between PD and autoimmune diseases such as inflammatory bowel disease, type 1 diabetes, and rheumatoid arthritis has also been seen in genome sequencing which could point to a shared susceptibility and the important role of the immune system in PD disease [51].

In epidemiological studies, certain environmental factors have consistently been found to be associated with an increased or decreased risk for PD. Pesticide exposure, rural living, and severe head trauma are associated with an increased risk for PD. On the other hand, lifestyle choices such as coffee drinking and cigarette smoking have been found to decrease the risk of PD [52]. In the context of neuroinflammation, some of these risk factors can be linked to immunotoxic effects, while in others links to neuroinflammation are more speculative. Pesticide exposure, pyrethroids, in particular, is linked to an increased risk of PD and immunotoxic effects have been observed in multiple mouse model studies [53]. Nicotine use is consistently shown to be associated with a lower risk of PD, which has also been observed in ulcerative colitis (UC). Ulcerative colitis is an autoimmune disease and nicotine has been shown to have anti-inflammatory properties through activation of α4/α7 nicotinic acetylcholine receptors on macrophages and a decrease of pro-inflammatory response as a result [54]. It can be hypothesized that similar anti-inflammatory properties could have an impact on neuroinflammation in PD.

As earlier stated, α-synuclein specific T-cell reactivity has been seen up to ten years before motor symptoms. α-synuclein protein is present in intestinal tissue such as mucosa of the appendix [55]. Some studies have shown higher levels of the toxic form of the protein in the gut in PD patients compared to controls [53].

## 3. Immunomodulatory Treatment Studies

There is no current consensus on an anti-inflammatory treatment of PD, but many candidates have been studied in the past. A 2011 Cochrane review analyzed non-steroid anti-inflammatory drugs (NSAID) as a preventive treatment for PD. Only observational studies regarding primary prevention were found at the time and the combined data showed a small reduction in risk of PD by regular use of ibuprofen [56]. Other NSAIDs were not shown to have any statistical effect on the risk of developing PD. In conclusion, there was not enough evidence to recommend any NSAID as a preventive measure, but it highlights the possible role of anti-inflammatory drugs as future treatments.

Some epidemiological studies have shown a lower risk of PD in patients who are taking other forms of immunosuppressive drugs. A population-based case-control study with around 50,000 patients and controls in the US showed a lower risk of PD in patients taking monophosphate dehydrogenase inhibitors or corticosteroids [57]. A retrospective cohort study analyzing the incidence of PD among patients with inflammatory bowel disease showed a higher incidence of PD among patients with IBD than without but also that early exposure to anti-TNF-α therapy was associated with a reduced PD incidence. This can be seen as evidence for the role of autoimmunity and also the possible role of immunosuppressive treatment in preventing or treating PD [45].

Drugs affecting microglia in PD have been studied by Jucaite and colleagues, who showed the drug candidate AZD3241 to reduce microglial activity by inhibiting the enzyme myeloperoxidase [17]. The trial was limited to 8 weeks and thus the study had no clinical rating reflecting progression as a primary endpoint. AZD3241 has not been further studied on PD patients but there is an ongoing trial (NCT04616456) studying the effects on patients suffering from multiple system atrophy (MSA) with the drug Verdiperstat, which is another name for AZD3241.

Vinpocetine has been suggested as a neuroprotective agent due to its effects on inflammatory signaling and cerebral blood flow [58]. In a randomized controlled trial on PD patients, vinpocetine reduced circulating mRNA levels of TLR2/4 and downstream inflammatory cytokines such as TNF-α. A small improvement in cognitive function was noted but no change in motor symptoms was seen [59]. The study was limited by only 14 days of treatment which in part explains the lack of change in motor symptoms.

Niacin (vitamin B3) has been proposed as a neuroprotective and anti-inflammatory treatment for PD because of its interaction with the G-protein-coupled receptor GPR109A. The receptor is expressed in monocytes, leukocytes, neutrophils, and macrophages and becomes upregulated in a pro-inflammatory environment [60]. Inhibition of GPR109A by niacin leads to decreased translocation of NF-κB and reduction of inflammatory cytokines in an in vitro study [61]. A case report with low-dose niacin treatment showed promising effects on PD symptoms and a randomized controlled trial (NCT03462680) has been completed although no data have been published as of yet [62].

Another ongoing trial by Greenland and colleagues (EudraCT-2018-003089-14) is studying the efficacy of the immunosuppressive drug azathioprine in a phase II randomized controlled trial with clinical progression as the primary endpoint [63].

## 4. Leukotriene Signaling Pathway

### 4.1. Leukotrienes

Leukotrienes are along with prostaglandins, lipoxins, and thromboxanes included in a group of long-chain fatty acids known as eicosanoids. They are known to play important parts in the inflammatory response such as leukocyte chemotaxis, vascular leakage, and astrocyte proliferation, and were first described by Bengt Samuelsson and colleagues in 1983 [64]. Leukotrienes (LT) are synthesized from free arachidonic acid (AA) by the enzyme 5-lipoxygenase (5-LOX) into LTA_4_, which is then further metabolized into LTB_4_, C_4_, D_4_, and E_4_ [65]. LTC_4_, D_4_, and E_4_ are grouped by their molecule structure to form the cysteinyl leukotrienes and they mainly activate two receptors, CysLT_1_ and CysLT_2_. CysLT_1_ is a Gq/11 family G-protein-coupled receptor with signaling through phospholipase C and Ca^2+^ mobilization [66]. The abundance of research on CysLTs has been done in the context of asthma where CysLT_1_ has been repeatedly shown to damage lung tissue through smooth muscle contraction and eosinophil migration [66,67]. CysLT receptors have also been implicated in cardiovascular disease, where CysLT signaling has been shown to induce vasoconstriction and chronic inflammation. Chronic inflammation is believed to play a main role in atherosclerosis [68].

### 4.2. Leukotrienes in CNS Disorders

In the CNS, evidence for CysLT_1_ expression has been seen in multiple cell types including neurons, endothelial cells, astrocytes, and microglia in several brain regions including cortex, hippocampus, striatum, and substantia nigra [69]. In the healthy brain, expression of CysLT receptors is low but several studies have shown that cerebral insult such as stroke or trauma increases the expression significantly [70,71,72]. In the context of brain injury, CysLTs have also been shown to increase blood-brain barrier permeability [73].

Expression of CysLT receptors has also been shown to be elevated in an MPTP-induced PD mouse model [74]. Specific upregulation of CysLT_1_ on microglia has been seen after rotenone stimulation in a BV2 microglial cell model suggesting an increased receptor expression in activated microglia [75]. The binding of CysLTs to microglial CysLT_1_ leads to an increased inflammatory response by upregulation of NF-κB through mitogen-activated protein kinase (MAPK), which ultimately leads to increased secretion of cytokines such as IL-1β and TNF-α. CysLTs also affect the phagocytosis of microglia and the production of free radicals through the regulation of nitric oxide synthase [76].

In AD, both in vivo and in vitro studies have shown that LTD_4_-induced upregulation of the CysLT_1_ receptor is correlated with increased β-amyloid and cognitive dysfunction in mice [77]. LTD_4_ directly upregulates beta-secretase (BACE)1, the enzyme responsible for β-amyloid production in neurons [78]. Similar expression of CysLT receptors on astrocytes and microglia is observed in AD as in PD mouse models, but upregulation of these cells is observed in the hippocampus and cortex where AD pathology is most abundant [79].

### 4.3. 5-Lipoxygenase

5-LOX is a key enzyme in the leukotriene pathway since it catalyzes the formation of LTA4 from AA [80]. 5-lipoxygenase is expressed in neuronal and glial cells providing a supply of LTA_4_ but evidence shows that LTA_4_ can also be supplied by peripheral neutrophils for further metabolization in brain resident cells [81].

In PD, 5-LOX has been shown to be involved in DA cell death in mice after MPTP injection. Knockout of 5-LOX showed a protective effect with no significant reduction in DA neurons after MPTP injection [82]. Similar results were seen in another study where the use of a 5-LOX inhibitor protected against neuronal cell death in both a human DA cell line and mouse model [83].

In AD, 5-LOX is upregulated in postmortem brains, and in vitro studies have shown 5-LOX overexpression to be linked with β-amyloid formation and tau hyperphosphorylation [84]. Studies on 5-LOX inhibitors such as zileuton have shown promising results in ameliorating neuroinflammation and improving neurogenesis in the hippocampus, which makes it a promising candidate for studies on other CNS disorders [85].

## 5. Montelukast

### 5.1. Discovery and Current Use

Pre-clinical research on CysLT receptors and their role in asthma pathology led to the development of the CysLT_1_ antagonist montelukast in the late 1990s [86,87]. Montelukast and other CysLT_1_ antagonists are currently used as an adjuvant therapy for children and adults suffering from asthma [88]. Other indications have been proposed and most evidence exists for the use of montelukast to treat allergic rhinitis, but its use in other inflammatory conditions, such as atherosclerosis and chronic obstructive pulmonary disease, have also been suggested [66,68]. Because of montelukast’s effect on inflammation and vascular damage in the lung, it has also been proposed as a treatment for COVID-19 associated pneumonitis [89].

### 5.2. Montelukast as a Treatment for PD

In a study from 2017, Jang and colleagues showed that montelukast protected DA neurons against microglial activation and attenuated the production of neurotoxic cytokines such as TNF-α and IL-1β in a mouse model of PD [90]. Similar results were seen in a study by Nagarajan and colleagues in 2018 where montelukast attenuated rotenone-induced microglial activation and had a protective effect on motor function deterioration [91]. A more detailed study on the diverse effects of montelukast in a PD mouse model by Mansour and colleagues showed similar results regarding reduced microglial activation and improved motor symptoms. Furthermore, a positive effect on neuronal survival was seen due to a downregulation of pro-apoptotic p53 expression and attenuated oxidative stress due to the ROS scavenging properties of montelukast (Figure 1) [92].

The authors of this review are currently recruiting patients for a phase II unblinded study on the safety and tolerability of montelukast in PD patients (EudraCT: 2020-000148-76). All participants will receive 40 mg of montelukast daily and microglial activity will be measured with PET imaging before and after 12-week treatment. Inflammatory biomarkers will be collected along with clinical rating scales.

### 5.3. Montelukast as a Treatment for other CNS Disorders

Montelukast treatment has been studied in AD animal models where it has been shown to have a rescuing effect of β-amyloid induced neurotoxicity and reversal of CysLT_1_R expression with a reduction of pro-inflammatory factors and apoptosis-related proteins as a result [93].

Montelukast has also been studied in the context of aging where its positive effect on microglial activation was again seen, but more interestingly, hippocampal neurogenesis was increased, which suggests that the drug could restore neuronal circuitries [94]. In a more recent study by the same authors, montelukast treatment was shown to reduce α-synuclein load and restore memory in an α-synuclein-based mouse model of Lewy-body dementia [95].

Building on the idea of montelukast as a treatment for cognitive impairment, a case study with 17 patients in 2017 showed promising subjective improvements in memory and other symptoms relating to dementia [96]. This hypothesis was tested in a register-based study in Norway where the possible effects of montelukast on neurological aging were analyzed. Data from the Norwegian prescription database and a prospective cohort study with health data from 45,000 people between 1974‒2016 was used. Results showed a correlation between the previous use of montelukast and improved scores on cognitive and neurological function tests [97].

There are also two ongoing phase II placebo-controlled trials on montelukast in Alzheimer’s disease (NCT03991988) and (NCT03402503). In the former, 15 participants are being recruited for a one-year study with 40 mg of montelukast or placebo and in the latter, a new buccal film formulation of the drug is being tested on 70 participants for 26 weeks.

An increase in CysLTs and CysLT receptors has been studied in the context of traumatic brain injury (TBI) where inflammation and disruption of the blood-brain barrier occur after a traumatic event [77]. The use of montelukast has been proposed as a treatment of TBI to attenuate chronic neurological damage caused by neuroinflammation [98].

In summary, the effects of montelukast on neuroinflammation in multiple neurodegenerative disorders are promising in animal models, where effects on both symptoms and disease biomarkers have been seen. Results from ongoing human trials will determine the possible role of montelukast as a treatment for inflammation in CNS disorders.

## 6. Conclusions

Parkinson’s disease is a complex and likely multifactorial disease. The exact cause is still unknown but emerging evidence points towards the immune system as an important driver for disease progression. As discussed in this review, the role of microglia has been the focus of a lot of research in recent years and its function as the main immune cell in the CNS makes it a reasonable treatment target. Earlier in vitro and animal studies have shown positive effects of microglia suppression on DA loss and PD symptoms, but larger human trials are scarce. Therefore, more pre-clinical and clinical research is needed to determine what parts of the CNS immune response can be targeted for treatment and at what stage of the disease these treatments would be the most effective. It is possible that more than one type of immune suppressant will be necessary to have an effect on disease progression. Hopefully, the results of ongoing clinical studies will shed some light on the topic and give rise to better treatment options for PD patients in the future.

## Figures and Tables

**Figure 1 ijms-22-05606-f001:**
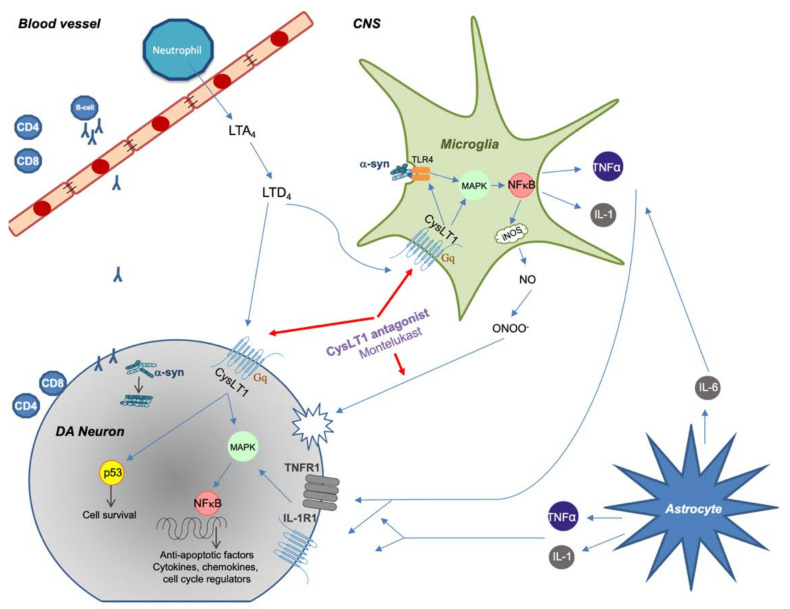
Summary of proposed immune system involvement in Parkinson’s disease (PD) with an emphasis on CysLT signaling. Peripheral immune cells are activated and can enter the CNS during pathological conditions. Misfolded α-synuclein in the CNS or periphery can trigger CD4 and CD8 T-cell recruitment with direct neuronal cell death as a consequence. α-synuclein can also activate microglia through TLR4-mediated phagocytosis, which causes an increased inflammatory response through NF-κB signaling. Astrocytes react to tissue damage and further enhance inflammatory response through production of cytokines. CysLTs increases inflammatory response by binding to CysLT_1_ receptor, thus further increasing microglial activity. CysLT_1_ on DA neurons can trigger apoptosis by upregulating p53 gene. Montelukast decreases inflammation by inhibiting CysLT_1_ signaling but can also act as an antioxidant by binding directly to reactive nitrogen species.

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
