# Peer review of "Potential Effects of Leukotriene Receptor Antagonist Montelukast in Treatment of Neuroinflammation in Parkinson’s Disease"

_ijms, 2021, doi:10.3390/ijms22115606_

Round 1
Reviewer 1 Report
The submission from Johan Wallin et al. reports the potential effects of leukotriene receptor antagonist Montelukast in treatment of neuroinflammation in Parkinson’s disease. The review is interesting and well structured. However, there are some corrections.
Minor comments:
- Line 65: The authors should add the type of TNF
- The authors should add some more references about Parkinson’s disease (DOI: 1371/journal.pone.0174470; DOI: 10.1007/s12035-018-1064-2; DOI: 10.1007/s12035-017-0496-4) and the role of TLR4 (DOI: 10.1016/j.bbi.2018.12.003
- I suggest that the authors dedicate more sections to Montelukast and the pathway of leukotrienes, or to change the title.
- The authors should better check the manuscript for any typographical errors.

Reviewer 2 Report
This is a very interesting and informative review. The authors provide the information necessary to understand why inhibitors of microglia activation using CysLT1 antagonists may be helpful in PD patients. The authors themselves are recruiting for clinical trials using Montelukast in Parkinson's disease (PD) patients. Therefore, this review provides an interesting window in the rationale of testing whether this drug may be useful for cognitive problems in PD.
I have only minor comments.
The text has few grammatical mismatches of number, combining plurals and singular throughout the text also please double check the text to avoid compositions like: "an phase" in line 242.
Please add a brief paragraph indicating the pharmacological differences between 11C-PK11195 and 11C-PBR28, and whether those differences may explain the difference in results.
Please provide a definition of abbreviations at its first use, or provide the definitions as a footnote (e.g., MPTP, GPR109A). In the particular case of the receptors please also add a brief description of its principal function.
Provide the primary reference for the presence of alpha-syn in the appendiceal mucosa.
